**Data Availability Statement:** All relevant data are within the paper.

# Premature cognitive decline in specific domains found in young veterans with mTBI coincide with elder normative scores and advanced-age subjects with early-stage Parkinson's disease

**Vicki A. Nejtek**[1]☯*, **Rachael N. James**[1]☯, **Michael F. Salvatore**[1]☯, **Helene M. Alphonso**[1,2‡], **Gary W. Boehm**[3‡]

1 University of North Texas Health Science Center, Fort Worth, Texas, United States of America, 2 John Peter Smith Health Network, Fort Worth, Texas, United States of America, 3 Texas Christian University, Fort Worth, Texas, United States of America

☯ These authors contributed equally to this work.
‡ HMA and GWB also contributed equally to this work.
* vicki.nejtek@unthsc.edu

## Abstract

### Importance

Epidemiologists report a 56% increased risk of veterans with (+) mild traumatic brain injury (mTBI) developing Parkinson's disease (PD) within 12-years post-injury. The most relevant contributors to this high risk of PD in veterans (+) mTBI is unknown. As cognitive problems often precede PD diagnosis, identifying specific domains most involved with mTBI-related PD onset is critical.

### Objectives

To discern which cognitive domains underlie the mTBI-PD risk relationship proposed in epidemiology studies.

### Design and setting

This exploratory match-controlled, cross-sectional study was conducted in a medical school laboratory from 2017–2020.

### Participants

Age- and IQ-matched veterans with (+) and without mTBI, non-veteran healthy controls, and IQ-matched non-demented early-stage PD were compared. Chronic neurological, unremitted/debilitating diseases, disorders, dementia, and substance use among others were excluded.

**Funding:** VAN was the Principal Investigator funded, in part, by the Graham and Caroline Holloway Family Foundation, the JES Edwards Foundation under Grant # RP20007, and the Institute for Translational Research (formerly the Center for Alzheimer's and Neurodegenerative Disease Research) at UNT Health Science Center under Grant # RI10024. The funders had no role in study design, data collection and analysis, decision to publish, or preparation of the manuscript.

**Competing interests:** The authors have declared that no competing interests exist.

## Exposure

Veterans were or were not exposed to non-penetrating combat-related mTBI occurring within the past 7-years. No other groups had recent military service or mTBI.

## Main outcomes / measures

Cognitive flexibility, attention, memory, visuospatial ability, and verbal fluency were examined with well-known standardized neuropsychological assessments.

## Results

Out of 200 volunteers, 114 provided evaluable data. Groups significantly differed on cognitive tests [$F(21,299) = 3.09$, $p<0.0001$]. Post hoc tests showed veterans (+) mTBI performed significantly worse than matched-control groups on four out of eight cognitive tests (range: $p = .009$ to $.049$), and more often than not performed comparably to early-stage PD (range: $p = .749$ to $.140$).

## Conclusions and relevance

We found subtle, premature cognitive decline occurring in very specific cognitive domains in veterans (+) mTBI that would typically be overlooked in a clinic setting, This result potentially puts them at-risk for continual cognitive decline that may portend to the eventual onset of PD or some other neurodegenerative disease.

## Introduction

Epidemiologists report that veterans (+) mTBI have a 56% increased risk of developing idiopathic Parkinson's disease (PD) within 12-years post-injury [1]. This finding is critically important as over 82% of the ~430,000 head injuries suffered by active-duty servicemen and women from 2000–2018 were classified as mTBI [2]. Operation Enduring Freedom, Operation Iraqi Freedom, and Operation New Dawn veterans have suffered the highest number of non-penetrating mTBI compared to soldiers involved in any other previous conflict [3, 4]. In some veterans, exposure to mTBI has resulted in cognitive impairment evident long after the injury. Specifically, Vanderploeg et al. (2004) [5] found impaired complex attention and working memory in veterans (+) mTBI ~8-years post-injury; however, these particular data were not examined as a potential link to PD onset.

Several other studies have reported that a significant proportion of concussed individuals experience residual cognitive problems evident 1- to 10-years post-injury [1, 6–12]. Again, these studies did not intend to examine post-concussion cognitive problems as risks for, links to, or in comparison to cognitive functioning in PD. Thus, while residual cognitive difficulties may hint at a mTBI-related risk for PD, to our knowledge there is little, if any, prospective evidence to characterize the earliest cognitive manifestations of mTBI in a young group of individuals that might be even remotely similar to those with PD. Prior to further considering that mTBI might be a risk for PD based on epidemiological studies, it seems clinically relevant to advance precision medicine by first trying to identify potential relationships that might link mTBI to PD. One way to investigate this possible link could be to compare cognitive functioning in older, early-stage PD to younger concussed individuals such as veterans (+) mTBI.

For example, in the preclinical/premotor phase of PD, memory problems, inattention, and decreased cognitive flexibility occur in up to ~24% of individuals [13]. Moreover, Durcan et al. (2019) [14] found 36% of PD patients experience memory and attention problems ~39-months prior to motor decline. Other data indicates that cognitive problems occurring 7- to 13-years prior to frank motor impairment can actually be considered a prodromal marker of Parkinsonism and probable PD [15]. Thus, it appears that there may be specific domains of cognitive function that are compromised prior to motor decline in PD. Whether or not a cognitive phenotype could link mTBI to PD requires investigation. So far, the literature has many ambiguities in establishing a cognitive pre-motor phenotype to identify the mTBI-related risk for PD such as: (1) inconsistent retrospective data from large electronic databases, the (2) intermingling of various PD stages or TBI severity levels, and until now (3) no matched-controlled investigations [16–19]. Moreover, there is no consensus about the specific cognitive problems that are common to both mTBI and PD that might be useful in a mTBI-PD prediction model. These ambiguities in the literature make it very difficult to identify a particular domain of cognitive functioning that might be considered a prodromal phenotype of mTBI-related risks for PD. As such, the gap in our understanding about cognitive problems or premature cognitive decline that may link mTBI to future risks for PD is considerable.

Thus, identifying a mTBI-related cognitive characteristic common to both mTBI and PD might help close this gap. To that end, we conducted an exploratory, matched-control study to examine post-mTBI cognitive functioning in age- and IQ-matched young veterans (+) mTBI, non-mTBI veterans (hereafter simply referred to as veterans), and non-veteran healthy controls (hereafter referred to as healthy controls) against older, early-stage PD subjects also IQ-matched. The study was not designed to investigate clinically significant cognitive impairment, but to explore subclinical cognitive problems that would typically be overlooked in a clinic setting. Examining any similarities in cognitive abilities between young mTBI and older early-stage PD could provide much-needed evidence of a cognitive phenotype characterizing mTBI-related links to or risks for PD when used in conjunction with other prodromal PD characteristics (i.e. REM sleep disorder, hyposmia, etc.) [20].

## Methods and materials

### Study design and setting

The goal of the study was to obtain evidence for premature cognitive decline in young veterans (+) mTBI by comparing their cognitive scores against veterans without mTBI, healthy non-veteran controls, and older early-stage, non-demented subjects with PD. Here, an exploratory, matched-control, cross-sectional study was conducted matching for age (+/- 5-years) and IQ (minimum 90 points) in the veteran and healthy control groups while IQ in the PD group was matched with the younger groups. The study was conducted between 2017–2020 in a medical laboratory setting following STROBE guidelines.

### Participant recruitment

After University of North Texas Health Science Center Institutional Review Board approval was obtained in 2017, community-based participatory recruitment was conducted by reaching out to health professionals, local veteran and PD non-profit organizations, member associations, support groups, academic undergraduate / graduate student organizations, and study participant family members via word-of-mouth. Recruitment ads were also placed on our university daily news blast, social media websites, newsletters, and were hand-distributed at community-wide events. Each subject provided written informed consent prior to study enrollment.

## Inclusion / exclusion criteria

Eligible volunteers were men and women veterans, veterans (+) mTBI, non-veteran healthy controls (25-45-years-old), and early stage, non-demented PD subjects (60–90-years-old) of Black/African American, Non-Hispanic White and Hispanic descent with at least 12-years of education and scoring ≥ 90 points on the American National Adult Reading Test (AMNART) using the Grober and Sliwinski equation where pronunciation errors plus education years measures verbal IQ as an estimate of premorbid intelligence [21]. In addition to non-veteran healthy controls, non-mTBI veterans were recruited to serve as a military control group for the veterans (+) mTBI. Exclusion criteria were non-English speaking, pregnant, had any unremitted/debilitating endocrine, autoimmune, rheumatological, cardiovascular, psychiatric or neurological diseases / disorders (other than mTBI or early-stage PD), dementia, had any physical or sensory limitations preventing completion of study procedures, alcohol, illicit or legal substance use, or tobacco use that could influence cognitive test performance. All subjects were required to abstain from drinking caffeine and sugary drinks 1-hour prior to study visit. A urine drug screen, breathalyzer, and fasting blood draw were collected to identify excluded substances.

The study physician (H.A., board certified in psychiatry and neurology), verified the presence of mTBI, early-stage PD, and absence of excluded psychiatric and neurological brain diseases / disorders in all subjects using standard psychiatric and neurological diagnostic criteria (i. e. American Psychiatric Association; American Neurological Association). All subject data were reviewed by a consensus diagnostic group (Principal Investigator, study physician, and research assistants trained in psychiatric, neurological, and cognitive testing). The Mini Mental State Exam (MMSE; 24 minimum score) was used to exclude subjects with mild cognitive impairment and dementia [22, 23]. A brief interview was conducted in all subjects to collect demographic and general health information. The Mini Neuropsychological Interview Version 6.0 [24] was used to screen out suicide, current and unremitted mood and substance use disorders in all subjects. Veterans (+) mTBI were exposed within the past 7-years during their military career to non-penetrating mTBI. Healthy controls and PD subjects had no recent military service or mTBI histories. Further screening protocols for PD and mTBI are shown below.

## Parkinson's disease screening

All PD volunteers had an early-stage qualified diagnosis made by their private treating neurologist prior to entering the study. The study physician (HA) used the Hoehn and Yahr Scale [25, 26] and the Unified Parkinson's Disease Rating Scale (UPDRS) to verify early stage (1–1.5) criteria and to screen out later stages of PD (>18 on Part III motor section) in accordance with Movement Disorders Society Task Force 2003 recommended guidelines and other evidence-based data [27, 28]. Three subcategories in the UPDRS assess mentation, activities of daily living (ADL), and motor symptoms using a scale of 0–4 where '0' = None or Normal, '1' = Minimal or Mild, '2' = Mild, Some, or Moderate, and '3' = Severe, and '4' = Most severe. Part I Mentation has 4 questions and a score range of 0–16; Part II ADL has 13 questions and a score range of 0–52; and Part III Motor has 14 questions with a score range of 0–56. Mean scores for PD subjects for Part I Mentation = .63 +/- 1.03, Part II ADL = 3.4 +/-2.8, and Part III Motor = 9.07 +/- 4.4. All PD subjects met criteria for early stage and reported either None, Normal, Minimal or Mild PD symptoms indicating that they were well within the criteria for early-stage.

## Mild traumatic brain injury screening

To rule out more severe brain injury than concussion, the following criteria substantiating previous non-penetrating mTBI diagnosis were used: (1) loss of consciousness of ≤ 30-minutes,

(2) any loss of memory ≤ 24-hours for events immediately before or after the accident, (3) altered mental state feeling dazed, disoriented or confused after the accident ≤ 24-hours and/ or (4) scores on the Glasgow Coma Scale of 13–15 (if available) [9, 29–31]. Additionally, only the most pertinent questions from the Ohio State University TBI Identification Method Short Form [29], the 3 Question Brief Traumatic Brain Injury Screen used by the Defense and Veterans Brain Injury Center [9], and the Boston Assessment of TBI-Lifetime (BAT-L) assessment used by the VA Boston Healthcare System to evaluate OEF and OIF veterans [30] were combined into a single questionnaire to verify the presence of non-penetrating mTBI and to exclude those with more severe head trauma other than mTBI prior to study enrollment. Using these combined tools allowed researchers to collect: (1) current symptoms, (2) past symptoms associated with concussion Grade I, II, or III following Veteran Administration and Department of Defense consensus criteria, and (3) nature or origin of the injury. Specifically, the BAT-L [30] further described more details for considering concussion grading as Grade I = no loss of consciousness, <15-minutes of altered mental state or amnesia; Grade II = < 5-minutes loss of consciousness, > 15-minutes < 24-hours of altered mental state or amnesia; and Grade III = > 5-minutes < 30-minutes loss of consciousness, < 24-hours hours of altered mental state or amnesia.

## Posttraumatic stress and depression

Although all subjects were interviewed with the MINI Neuropsychological Interview to screen out current serious or unremitted mood disorders, we further evaluated all subjects with both the 17-item PTSD Checklist (PCL) [32] and Geriatric Depression Scale-30 (GDS) [33]. As mTBI is often associated with posttraumatic stress symptoms with potential to influence cognitive functioning, the publicly available PCL was used to determine stress severity levels [32]. Each item endorsing the severity of their problem associated with a particularly stressful event are scored as 1 = Not at all, 2 = A little bit, 3 = Moderately, 4 = Quite a bit, and 5 = Extremely. Each subject's PCL raw scores were summed and categorized as 17–35 = Little or no problems; 36–44 = Some to moderate problems, while 45–85 = Quite a bit to extreme severity of problems as cited by the National Center for PTSD www.ptsd.va.gov and Weathers et al.,1993) [32]. Accordingly, cut scores necessary to meet a PTSD diagnosis are 44 for general populations and 50 for military personnel. Raw mean scores for each subject group were used to describe the sample in relation to the cut scores. Depression often co-occurs with posttraumatic stress and levels of depression may also influence cognitive functioning. Thus, depression symptoms were also assessed with the GDS [33] from which summed raw scores were categorized into severity levels of normal (0–9), mild (10–19), or severe (20–30) as per Yesavage et al. (1983) [33].

## Cognitive outcomes

Standardized cognitive assessments used in this study (1) are listed in the National Institute of Neurological Disorders and Stroke Common Data Elements, (2) have been extensively used in neurodegenerative disease and aging research, and (3) have well-known reliability and validity coefficients [34–44]. The primary outcomes of interest were 'time-to-completion' total seconds on Trail-Making Test-A and -B (TMT-A, TMT-B) which were used to assess cognitive flexibility, attention, processing speed, memory, and inhibitory control [34, 37, 39–42]. The longer it takes to complete Trail-making tests, the poorer the cognitive functioning.

Secondary outcomes included: (a) total correct points (0–15 points for correct clock face construction showing the time 1:45) for CLOX-1 (constructed from memory) and CLOX-2 (copying a well-constructed clock face showing 1:45) which, both measure memory and

visuospatial abilities, (b) Wechsler Memory Scale (WMS) III evaluating forward (FWD) and backward (BWD) digit span recording the total correct sequence of digits recalled after hearing the correct sequence example primarily measuring working memory, (c) total number of correct words generated in sixty seconds for Controlled Oral Word Association Tests–F.A.S. assessing phonemic verbal fluency and Animal Naming test measuring semantic verbal fluency [34–38, 40, 43, 44]. The higher the total scores for these secondary outcomes, the better the cognitive functioning.

## Statistical analyses

Descriptive statistics were used to characterize the sample. Chi-square analyses tested potential distribution differences in gender and race/ethnicity among groups. One-way ANOVA was used to compare age and IQ across groups. Independent variables were Group and PCL severity levels. Dependent variables were total scores for each cognitive test. As symptoms of post-traumatic stress and depression are known to overlap, we examined the relationship between PCL and GDS with a *Pearson* correlation using raw scores. PCL severity levels were also analyzed as an influencing covariate on cognitive outcomes with multivariate analyses GDS severity levels.

Raw scores were used for statistical comparison analyses among the groups. Descriptive comparisons to normative scores were used to further characterize the sample's performance against the general population. Accounting for unequal group sample sizes, a General Linear Model (GLM) one-way MANOVA tested Group effects, and a 2-way MANOVA tested main effects and interactions of Group and PCL on cognitive outcomes. To control for multiple comparisons and to help draw statistical inferences, we used Benjamini and Hochberg (1995) [44] False Discovery Rate (FDR) post-hoc analyses (as appropriate) to limit false positive results, while shedding light on the cognitive tests that might reveal subtle differences in cognitive functioning among the groups [45, 46]. A 95% confidence interval and two-tailed alpha probability level of .05 was used to determine statistical significance using the Statistical Package for the Social Sciences, version 23 (IBM Corp., Armonk, N.Y., USA).

## Results

Out of 200 volunteers who provided written informed consent to undergo screening procedures, 46 did not meet inclusion/exclusion criteria (e. g. excluded disorder/disease, tested positive for an excluded substance, etc.); 24 were unavailable or unwilling to undergo full study procedures after screening; and 7 did not meet age- or IQ-matching criteria. This resulted in 123 volunteers who enrolled in the study. Out of 123 participants, there were 9 excluded outliers (2 healthy controls, 1 veteran, 3 veterans (+) mTBI, and 3 PD subjects) whose cognitive test scores were > 2 standard deviations either above or below the standardized normative test values or symptoms revealed during the MINI Neuropsychological interview that met clinical criteria for exclusion were discovered only after enrollment. The MMSE raw score range was 24–30 which indicated that none of the remaining 114 subjects had dementia or clinically-relevant cognitive impairment [47]. Mean scores on the MMSE for each group were: Veterans (+) mTBI = 28.9 +/- 1.3; Veterans = 29.0 +/- 1.2; Healthy controls = 29.3 +/- 1.0; and PD group = 28.4 +/- 1.5 with the overall sample mean = 28.93 +/- 1.29. Group means and standard deviations from the AMNART [21] test showed the sample had above average estimated premorbid IQ: Veterans (+) mTBI = 115.4 +/- 6.7; Veterans = 117.50 +/- 5.8; Healthy controls = 119.5 +/- 6.3; and PD group = 117.6 +/- 6.5 with the overall sample mean = 117.6 +/- 6.5.

## Demographic, clinical, and military characteristics

One-hundred-fourteen subjects (35 women, 79 men; 70 Non-Hispanic White, 29 Hispanic, 15 Black/African American) provided evaluable data consisting of 27 veterans (+) mTBI (M = 33.4, SD = 5.5-yrs. old), 30 veterans (M = 32.7, SD = 3.8 yrs. old), 30 healthy controls (M = 31, SD = 4.9 yrs. old), and 27 early-stage PD subjects (M = 68.5, SD = 8.3 yrs. old). Among veterans (+), veterans, and healthy controls there were no significant differences in age [$F(2,84)$ = 1.83, $p$ = .167, CI = 31.37–33.43]. Among all four groups, there were no significant differences in IQ [$F(3,110)$ = 1.93, $p$ = .129, CI = 116.42–118.83], gender [$\chi^2$ (3, $N$ = 114) = 3.68, $p$ = .298], or race/ethnicity [$\chi^2$ (6, $N$ = 114) = 8.75, $p$ = .188].

The mean number of years of military service was 9.04 +/- 5.6 for veterans (+) mTBI and 8.13 +/- 3.9 for non-mTBI veterans. In the mTBI veteran group, 12 served in the Army, 9 were Marines, 3 served in the Navy and 3 were in the Air Force. Of the non-mTBI veteran group, 11 served in the Army, 8 were Marines, 6 served in the Air Force, 4 were in the Navy, and 1 served in the National Guard. The most prevalent deployment location was Iraq and Afghanistan for both veterans (+) mTBI (n = 18) and non-mTBI veterans (n = 19). The remaining deployments varied among the Persian Gulf, Saudi Arabia, Haiti, South Korea, Haiti, Asia, and South America.

All mTBIs were non-penetrating injuries occurring within the past 7-years (M = 5.1, SD = 1.9) that resulted from military-related blasts/explosions (n = 9), motorized accidents (n = 7), physical assaults (n = 6), or falls (n = 5). The mean number of mTBIs experienced throughout military service was 3.6 +/- 2.8. The BAT-L concussion grading criteria showed a majority of Grade I (n = 13) and Grade II (n = 10) and a few Grade III (n = 4) concussions [30]. Current post-mTBI related symptoms of tinnitus (n = 15), headache (n = 15), and irritability (n = 13) were the most commonly reported overlapping symptoms, while others reported light sensitivity (n = 7) and blurred vision (n = 5).

For PD subjects, two were levodopa naïve (de novo) and all other PD subjects were tested in the 'ON' medication state. Twelve PD subjects received below the 550 mg/day median levodopa equivalent doses while 13 received above the 550 mg/day median dose ($M$ = 630 mg/day, $SE$ = 89.7, 95% C.I. = 445.9, 814.6). There was no influence of medication on TMT-A [$F(1,25)$ = .01, $p$ = .92], TMT-B [$F(1,25)$ = .12, $p$ = .73], CLOX1 [$F(1,25)$ = 3.26, $p$ = .083], CLOX2 [$F(1,25)$ = .027, $p$ = .87], WMS III number of correct total FWD + BWD [$F(1,25)$ = 2.35, $p$ = .132], FAS [$F(1,25)$ = .004, $p$ = .95], or Animal Naming [$F(1,25)$ = .09, $p$ = .77].

## Posttraumatic stress and depression

Posttraumatic stress and depression were evaluated using the PCL and the GDS in keeping with the tools used in Alzheimer's and aging research. To our knowledge there are no studies validating PCL use in early-stage PD and none validating the GDS use in young veterans (+) mTBI. For the PCL, the raw mean score for healthy controls (M = 20.5 +/-4.5), PD subjects (M = 19.3 +/- 3.1), veterans (M = 30.7 +/- 12.6), and veterans (+) mTBI (M = 38.2 +/- 13.8) indicate that these scores are below the diagnostic cut score of 50 for military personnel and 44 for the general population [32; www.ptsd.va.gov]. While veterans and veterans (+) mTBI reported higher stress levels than the other two groups ($\chi^2$ (6, $N$ = 114) = 41.07, $p$< .001) this result is expected. What is more informative is to realize that this sample did not meet DSM-IV criteria for a clinical diagnosis of PTSD according to the literature for military personnel. Further, we found no interaction effect of PCL severity levels influencing cognitive test results [$F$ (21, 285) = .897, $p$ = .596; Wilks' λ = .832, partial $\eta^2$ = .059].

Mean scores from the GDS showed healthy controls (M = 2.73 +/- 2.6), veterans (M = 5.93 +/- 5.1), PD subjects (M = 4.04 +/- 3.0), and veterans (+) mTBI (M = 8.3 +/- 6.3) were within

the normal (0–9) range. Although these mean scores were within normal limits, depression and posttraumatic stress are known to overlap. Therefore, we examined the possibility of a relationship between GDS and PCL and found a significant correlation ($r$ (112) = .654, $p < 0.001$). Based on the above results illustrating all groups had GDS scores within the normal range and the high degree of multicollinearity found between PCL and GDS, no further analyses were conducted using GDS or PCL scores.

## Cognitive outcomes

Multivariate analyses showed that there was a significant effect of Group across the different cognitive tests, [$F$(21, 299) = 3.09, $p < 0.0005$; Wilks' λ = .569, partial $\eta^2$ = .172]. Five out of eight tests significantly differed which suggests that at least some domains of cognitive functioning in the remaining tests may either be unaffected by mTBI or PD, perhaps the tests themselves lacked the sensitivity to discern subclinical cognitive problems. The non-significant tests were WMS-III (p = .094) [FWD (p = .263) + BWD (p = .090)], and FAS (p = .481) as digit span and verbal fluency scores were within normal limits for all groups and are therefore not areas of concern in our sample.

Fig 1 illustrates statistical differences and similarities for our primary outcomes (TMT-A and TMT-B) 'time to complete' raw means among the Groups. Veterans (+) mTBI performed more similarly to PD subjects than their matched controls and also performed as if they were decades older according to the normative data. The tests showing significant group differences were TMT- A [$F$(3, 110) = 10.64, $p < 0.0001$, partial $\eta^2$ = .225], TMT-B [$F$(3, 110) = 14.78, p < 0.0001, partial $\eta^2$ = .287], CLOX 1 [$F$(3, 110) = 3.60, $p$ = 0.016, partial $\eta^2$ = .089], CLOX 2 [$F$(3, 110) = 3.49, $p$ = 0.018, partial $\eta^2$ = .087], and Animal Naming [$F$(3,110) = 3.76, $p$ = 0.013, partial $\eta^2$ = .093]. Although there were statistical differences using the global '$F$', these differences are not informative nor meaningful until the post hoc results are examined. The Benjamini-Hochberg FDR [45, 46] post-hoc analyses showed the following results:

1. There were no significant performance differences between veterans *vs*. healthy controls on TMT-A ($p$ = 1.00) or TMT-B ($p$ = 1.00) indicating that both groups adequately served as controls for veterans (+) mTBI.

2. Veterans (+) mTBI took significantly longer to complete TMT-A and TMT-B compared to veterans (TMT-A: $p$ = .007; TMT-B: $p$ = .013), and healthy controls (TMT-A: $p$ = .014; TMT-B: $p$ = .006).

3. There were no statistical differences between veterans (+) mTBI and the PD group for TMT-A ($p$ = .749) or TMT-B ($p$ = .140) suggesting that these groups performed similarly on tests measuring cognitive flexibility, attention, processing speed, and memory (Fig 1).

4. The only significant difference found with CLOX 1 scores was between PD *vs*. healthy controls ($p$ = .009). Keeping in mind that the healthy controls were age-matched to serve as controls for both veteran groups, this result is not surprising. The large difference between these two groups were obviously driving the global 'F' and does not help establish a link from mTBI-related cognitive problems to problems also common in early-stage PD.

5. The only significant difference found in CLOX 2 scores was between veterans (+) mTBI *vs*. healthy controls ($p$ = .049). Veterans (+) mTBI, veterans, and PD subjects did not statistically differ on this visuospatial memory test. With all of the controls in place with this study design, it is unclear why veterans did not perform this test more like healthy controls.

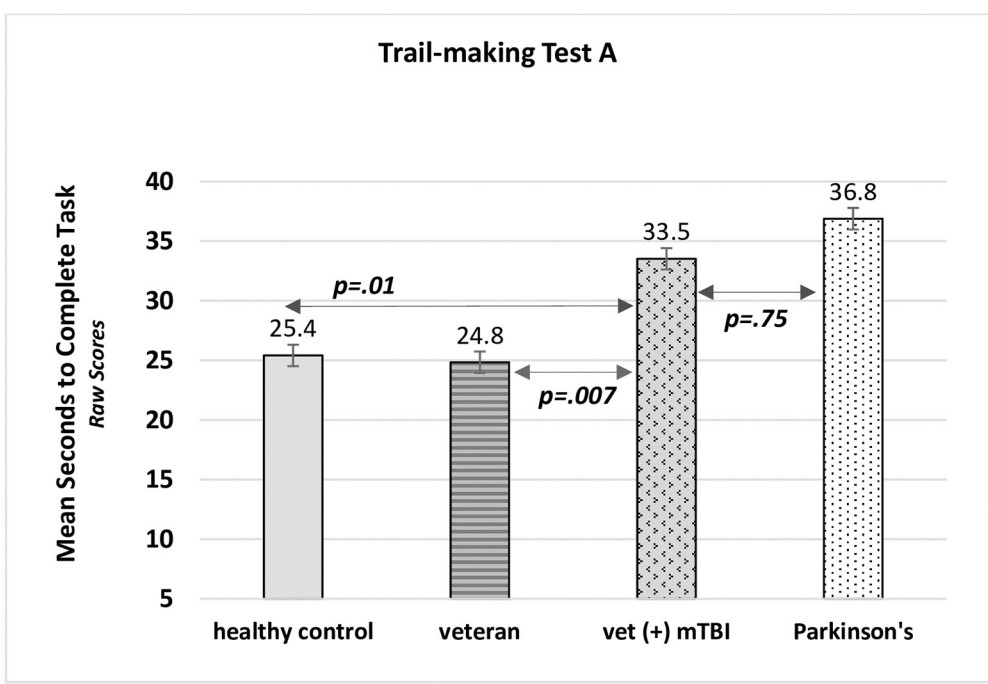

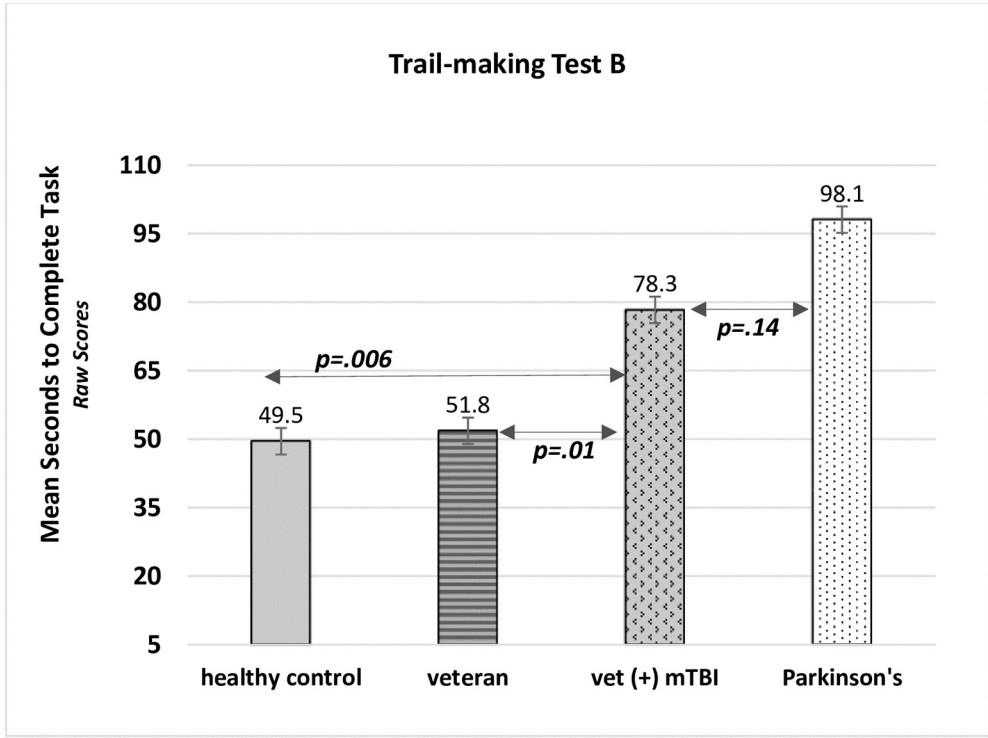

**Fig 1. Trail-making A and B reveal group differences and similarities in cognitive flexibility.**

6. Similarly, the only significant difference in Animal Naming was between veterans (+) mTBI *vs.* healthy controls ($p = .015$) although veterans (+) mTBI recalled fewer animals than any other group. Interestingly, PD subject performance reflected the 80th percentile using normative age and education stratification data. This result describes PD subjects as

having high average cognitive functioning [34, 37, 38, 40]. All things considered PD subjects outperformed all other groups on this test of semantic verbal fluency.

### Normative score interpretation

Normative scores provided objective evidence from which illustrative comparisons between study subjects and the general population can be inferred. Generating percentiles from cognitive scores derived from our small sample of veterans, mTBI, healthy controls, and PD subjects would not inform us about premorbid executive functioning compared to the average person. Instead, it is more informative to use standardized normative percentiles or normative mean scores stratified by age and education (as available) to characterize our sample's cognitive performance relative to the general population. We found available age- and education-based normative means and percentiles for TMT-A, TMT-B, FAS, and Animal Naming, while normative percentiles for CLOX 1 and 2, FWD, and BWD were unavailable. The available normative data for CLOX 1, CLOX 2, FWD, and BWD used in this study were age- and education-matched means and standard deviations to illustrate each group's cognitive test performance in relation to other general populations.

Table 1 provides age and education stratified normative percentiles (when available) or normative means derived from a general population for side-by-side descriptive comparisons corresponding to the raw means, standard deviations, and confidence intervals for each cognitive test and group. Group raw mean scores for healthy controls and veterans were within the 40–50th normative percentile for TMT-A, TMT-B, and FAS while Animal Naming scores were in the 75th percentile. These percentiles represent average performance on Trail-making and high average on semantic verbal fluency [34, 37–42].

Veterans (+) mTBI generally performed at a lower percentile than study controls (veterans and healthy controls) on four out of eight tests. For TMT-B results, veterans (+) mTBI took 26.5 to 28.8 seconds longer to complete than age- and IQ-matched veterans and healthy controls, respectively. What was totally unexpected was the extent of decline observed with TMT-B raw mean scores that corresponded to $< 10^{th}$ percentile indicating below average cognitive functioning in veterans (+) mTBI with response times of 11.2 seconds longer than their age- and education normative mean. This group performed akin to 60–64-year-old healthy individuals representing the 15th percentile, 65–69-year-olds at the $< 10^{th}$ percentile, or 70–74-year-olds whose means are representative of the 60th percentile [37, 39–42].

While not as obvious as the TMT-B outcome, veterans (+) mTBI performed the TMT-A in the 20th percentile indicating low average performance [39, 42]. Here, TMT-A mean scores correspond to the age and education normative mean for 55–59-year-olds scoring in the 30th percentile or 70-74-year-olds scoring in the 60th percentile [37, 39–42]. PD subjects also had raw means representing less than the 10th percentile indicating below average cognitive functioning on TMT-B [37, 39–42]. Here, PD subjects took up to 31 seconds longer to complete than the 10th percentile for their age- and education-based normative data. Thus, declining cognitive flexibility, attention, inhibitory control, processing speed, and working memory appear to be occurring in early-stage PD and in veterans (+) mTBI. Comparing these results to normative data in older populations have also helped to establish the characterization of premature cognitive decline in veterans (+) mTBI in very specific domains of cognitive functioning.

### Discussion

Some epidemiological data suggest that mTBI sets a risk trajectory for PD onset [1, 16, 18]. However, considerable debate remains about this assertion continue due to varied outcomes

**Table 1. Illustration of standardized, normative test percentiles or means stratified by age- and education corresponding to study group raw means.**

| Cognitive Tests (Scoring Parameter) | Normative Percentile Stratified by Age & Education Corresponding to Group Raw Mean | HC n = 30 Raw Mean (SD) | 95% C.I. | Veteran n = 30 Raw Mean (SD) | 95% C.I. | Vet (+) mTBI n = 27 Raw Mean (SD) | 95% C.I. | Normative Percentile Stratified by Age & Education Corresponding to Group Raw Mean | PD n = 27 Raw Mean (SD) | 95% C.I. |
|---|---|---|---|---|---|---|---|---|---|---|
| TMT-A [37, 39–42] (TTC-Seconds) | 40th%: HC, Veteran 20th%: Vet (+) | 25.4 (8.3) | 21.8–28.9 | 24.8 (7.3) | 21.3–28.4 | 33.5 (11.3) | 29.8–37.2 | 35th% | 36.8 (11.8) | 33.2–40.6 |
| TMT-B [37, 39–42] (TTC-Seconds) | 50th%: HC, Veteran <10th%: Vet (+) | 49.5 (12.2) | 37.9–61.1 | 51.8 (10.1) | 40.3–63.4 | 78.3 (29.8) | 66.1–90.5 | <10th% | 98.1 (56.2) | 85.9–110.3 |
| F.A.S. [34, 37, 38] (TW-1 minute) | 45th%: HC 35th%: Veteran 35th%: Vet (+) | 43.7 (10.8) | 40.0–47.4 | 39.8 (8.7) | 36.1–43.5 | 40.5 (8.2) | 36.6–44.4 | 50th% | 41.4 (10.2) | 37.5–45.3 |
| AN [34, 37, 38] (TW-1 minute) | 75th%: HC, Veteran 50th%: Vet (+) | 24.9 (4.1) | 23.2–26.7 | 23.3 (5.0) | 21.7–25.2 | 21.0 (3.8) | 19.2–22.9 | 80th% | 21.8 (5.9) | 20.0–23.7 |
|  | Normative Percentile Unavailable Normative Mean (SD) Shown Below |  |  |  |  |  |  | Normative Percentile Unavailable Normative Mean (SD) Shown Below |  |  |
| CLOX-1 [34, 36, 37, 43] (Total PCCC) | 13.2 (1.6) | 13.0 (.95) | 12.3–13.7 | 12.1 (1.8) | 11.5–12.8 | 12.1 (2.4) | 11.4–12.8 | 12.1 (2.6) | 11.4 (1.7) | 10.8–12.1 |
| CLOX-2 [34, 36, 37, 43] (Total PCCC) | 14.2 (1.2) | 13.9 (.66) | 13.5–14.3 | 13.3 (.98) | 12.9–13.6 | 13.2 (1.2) | 12.8–13.6 | 14.2 (1.0) | 13.2 (1.1) | 12.8–13.6 |
| FWD [35, 37, 39, 44] (Total CSD) | 6.7 (1.2) | 7.2 (1.2) | 6.8–7.7 | 6.8 (1.4) | 6.3–7.2 | 6.7 (1.2) | 6.3–7.2 | 6.4 (.88) | 6.6 (1.2) | 6.1–7.1 |
| BWD [35, 37, 39, 44] (Total CSD) | 5.0 (1.4) | 5.0 (1.2) | 4.6–5.5 | 4.6 (1.2) | 4.1–4.9 | 4.8 (1.3) | 4.3–5.2 | 5.0 (1.4) | 4.3 (1.0) | 3.9–4.8 |

HC = Healthy Controls; PD = Parkinson's Disease; SD = standard deviation; C.I. = Confidence Interval; TMT-A = Trail-making test A; TMT-B = Trail-making test B; AN = Animal Naming; FWD = Forward digit span; BWD = Backward digit span.

Scoring Parameters: TTC = Time to completion in seconds; TW = Total words generated in 1-minute; Total PCCC = Total Points Correct Clock Construction out of a possible 15; Total CSD = Total Correct Sequence of Digits recalled.

Note: Lower scores on TMT-A and TMT-B = better cognitive functioning. Higher scores on the remaining cognitive tests = better cognitive functioning.

Performance percentile description: >98th% (very superior); 91-97th% (superior); 75-90th% (high average); 25-74th% (average); 10-24th% (low average); 3-9th% (below average)

derived from an assortment of retrospective databases rather from controlled study designs [8, 17–19]. In this exploratory study we were searching for evidence to support the epidemiological studies inferring that mTBI initiates the eventual onset of PD. Although our study was not designed to predict risks for PD, we were able to show coincident similarities between veterans (+) mTBI and PD in particular cognitive domains. We also found veterans (+) mTBI significantly lagged behind their age- and IQ-matched controls, performed more like older, early-stage PD subjects and presented cognitively as if they were at least three decades older on tests of cognitive flexibility, attention, processing speed, and inhibitory control. Thus, veterans (+) mTBI show some premature cognitive decline, in very specific domains.

Other evidence shows veterans (+) mTBI had significantly more problems than healthy controls recalling animal names in a test of semantic verbal fluency, while other tests of phonemic verbal fluency and working memory associated with digit recall were unaffected. Further, veterans and veterans (+) mTBI performed visuospatial memory tests about the same as control groups. Thus, the differences we uncovered may signal a trajectory of eventual cognitive impairment in only very specific rather than global domains. These results indicate a differentiated outcome among the cognitive tests used to discern subclinical cognitive decline when using a complex test like Trail-making versus other types of tests like phonemic verbal fluency and working memory for digits.

In fact, the Trail-making test is one of the most sensitive tests to identify brain injury, severity of injury, and age-related cognitive decline in the domain of cognitive flexibility, sustained attention, processing speed, and response inhibition [12, 39–42]. Importantly, data suggest that longer than average time-to-completion scores for the TMT-A and TMT-B significantly predicts parkinsonism conversion to Lewy body dementia within 5–6-years [46]. Thus, TMT-A and TMT-B may be a powerful diagnostic tool to link m-TBI to PD, to infer future cognitive impairment, or the onset of some other type of neurodegenerative process [46]. However, it is also important to mention that even slight problems in performance on the Trail-making task might not be noticed if not compared to normative data from older individuals or matched controls as in the present study. Taken together, choosing specific tests that are sensitive enough to detect subtle mTBI-related cognitive problems could be used to benchmark an individual's performance as a premature cognitive aging phenotype with a potential trajectory of PD or some other neurodegenerative process.

For PD subjects, the data indicate that they are performing quite well for their disease state, age, and education in the domains of semantic and phonemic verbal fluency, working memory with digit sequencing, and visuospatial memory. In sharp contrast are other domains of cognitive flexibility, processing speed, and inhibitory control in which we found PD subjects functioning below average. Perhaps in this early stage of our PD subjects, decreased cognitive flexibility may be one of the first domains of executive functioning to falter signaling an imminent trajectory of clinically relevant impairment. In fact, two other matched-control studies who followed non-demented early-stage PD subjects across multiple years [46, 47] found abnormal TMT-B performance occurred 6-years prior to dementia onset and PD diagnosis [46], and another study reporting that TMT-B significantly differentiated those PD patients who were and were not at high risk for developing dementia over a 4.3-year time period [47]. These longitudinal study data highlight the specificity and sensitivity of TMT-B predicting progression of cognitive decline and the important role cognitive flexibility may play in predicting future risks for PD in veterans (+) mTBI. Understanding that phonemic verbal fluency, working memory, and visuospatial capabilities are relatively unaffected in early-stage PD and in younger post-mTBI adults is a positive finding that may inform therapeutic approaches to help offset problems these populations may have with cognitive flexibility.

Detecting subclinical cognitive changes in a clinic setting could provide the earliest evidence clinicians need to justify prescribing lifestyle and/or other non-pharmacological interventions to help mitigate the risks for PD in later years. Unfortunately, the typical cognitive test used in routine healthcare settings (e.g. Mini Mental State Exam) lacks the sensitivity to identify subclinical cognitive decline [48]. Moreover, young patients seeking medical care for routine wellness checks may not recognize or admit to cognitive complaints, or have no obvious symptoms. Thus, the potential for premature cognitive decline in young patients with a history of mTBI is likely to go unrecognized and untreated.

The same is true of cognitive problems heralding preclinical or prodromal PD as many seniors will relegate any cognitive loss to the generalities of 'old age'. Failure to recognize minor cognitive problems as an important harbinger of far worse things yet to come and a possible pre-curser of future neurodegeneration, begs for a reconsideration of mTBI and PD screening criteria currently used in clinical practice (e.g. MMSE) [48]. Expanding the use of non-invasive, easy to administer preclinical risk assessments to provide early identification of a mTBI-related or even non-mTBI-related link to PD, would give patients the best possible chance to improve their cognitive abilities prior to cognitive impairment, dementia, or motor decline.

Our results should encourage clinicians, patients, and caregivers alike to concentrate on nurturing and/or building cognitive reserve in the domains that seem unaffected while initiating therapeutic approaches to improve cognitive domains most at-risk for future decline. Studies suggest that a variety of non-pharmacological lifestyle interventions will effectively nurture and improve cognitive functioning for those with mTBI, PD, or any cognitively at-risk populations [49–53]. Those interventions include: playing word games, participating in daily aerobic activity, practicing meditation and mindfulness, learning new skills particularly in music, arts, and the theater, practicing healthy sleep hygiene, eliminating all stressors, partaking in nutritional meals, and establishing a strong social network [49–53].

Here, veterans (+) mTBI experienced their mTBI up to 7-years earlier and the subtle cognitive problems we found augments the accumulating evidence that shows residual cognitive problems are evident 8- to 12-years post-injury [1, 5, 12]. From the PD literature, recent evidence [15] shows cognitive problems occurring 7- to 13-years prior to motor impairment could be considered a prodromal marker for Parkinsonism and probable PD. Merging the mTBI and PD evidence could bring a fresh perspective towards recognizing a critical period of possible risks and eventual outcomes for both pathologies. Although the veterans (+) mTBI presented as highly functioning during their interview, their cognitive test results showed that they had difficulty with the specific cognitive domains as described previously. Whether or not performance on the tests used here could portend a risk for PD cannot be determined with the existing dataset, but should be further investigated along with other prodromal markers of PD such as REM sleep disorder, hyposmia, among others [20].

Aside from a potential link to PD risks, the specific cognitive domains identified as problematic indicates veterans (+) mTBI may have a distinct disadvantage in processing abstract concepts, distilling complicated information quickly, or paying attention to detail. Problems with information processing could potentially elicit a tendency to second-guess decisions, perseverate on unimportant details, or, conversely, make careless, non-strategic choices when attempting to make difficult decisions under duress [51–53]. Attention, cognitive flexibility, and processing speed influence response inhibition and are especially needed to ensure accurate solving of complex problems and strategic decision-making. These cognitive skills are necessary to adjust and adapt to the uncertainties of daily living, work demands, and social relationships [49, 52–54]. Particularly in military operations, hazardous situations and isolated work environments demand uncompromised executive functioning at a moment's notice

[53]. Similarly, in a PD population, these same cognitive skills are necessary to make strategic vs. careless decisions influencing therapeutic compliance *vs.* non-compliance impacting disease progression [49, 52, 54].

Evaluating mTBI and PD patients early and often no matter when their initial diagnosis occurred will allow clinicians an opportunity to detect cognitive decline as soon as possible. These results support the ultimate goal which is to promote detection in remotely concussed individuals and quickly initiate precision therapies to help delay, improve, or reverse progression of cognitive decline. Unrecognized and left untreated, subclinical cognitive decline may eventually convert to neurodegeneration with increased risks for impairment, dementia, PD, or some other disease process. Indeed, in non-demented early-stage PD, Pfeiffer et al., (2014) [55] found 70% of early-stage PD with no previous history of cognitive decline met criteria for a modified mild cognitive impairment diagnosis based on the outcomes from executive functioning/working memory, attention and concentration tests while other cognitive domains were relatively spared. This finding bolsters our previous suggestion that neurocognitive evaluations need to occur early and often for both post-mTBI injury and post-PD diagnosis patients in order to best identify the trajectory affecting the cognitive domains that are most vulnerable to progression of neurodegeneration and possible conversion to cognitive impairment or dementia.

## Limitations and strengths

The exploratory nature of this study was limited to a small sample size focusing on the cognitive problems experienced by veterans (+) mTBI and therefore may not be generalizable to a broader mTBI population. Another limitation of this study is the cross-sectional design which prevents us from longitudinally tracking the onset or progression of premature cognitive decline nor predicting PD risks in those with mTBI. Restricting criteria to early-stage PD might also be considered a limitation; however, allowing more advanced stages of PD would not allow us to examine the earliest cognitive problems in this patient group from which to compare veterans (+) mTBI. Moreover, more advanced stages of PD would have added enormous variability in functioning and in medication use. Although not assessing motor functioning in veterans (+) mTBI may be considered a limitation, our focus in this study was to first examine cognitive function (rather than motor impairment) as cognitive decline has been established as a prodromal PD phenotype that begins years prior to motor decline. Moreover, we excluded physical limitations that could prevent completion of study procedures and any kind of motor impairment could have inferred a more advance stage of PD than early-stage.

The preliminary data generated here represents an important first step in identifying specific cognitive domains that appear common to both mTBI and PD which may eventually help to characterize a cognitive phenotype from which a mTBI-related link to or risk for eventual PD can be made. Especially important was realizing that the MMSE scores, verbal phonetic fluency, working memory for digits failed to identify subtle cognitive problems, while TMT-A and particularly TMT-B were more sensitive in detecting premature decline in cognitive flexibility. Thus, other positive aspects of this study are identifying specific domains that contributed most to premature cognitive decline and narrowing down the domains that are most sensitive towards identifying subtle and subclinical cognitive problems in those with a remote history of mTBI and in early-stage PD.

Study strengths include our community-based participatory recruitment methods, the matched-control design, and the thorough use of structured interviews and standard diagnostic assessments that are consistent with existing age-related research paradigms. The study

design helped limit within- and between-group variance and our statistical methods controlled potential influencing variables such as depression and posttraumatic stress on cognitive outcomes. The extensive exclusion criteria such as IQ, mild cognitive impairment, comorbid mood and medical diagnoses, substance use, and medications known to influence cognition, also promoted quality control of the potential influence of other health-related variables potentially influencing cognitive performance.

## Conclusion

We present the first exploratory data of its kind describing a small, yet well-controlled comparison study providing compelling new evidence of subtle, subclinical problems in specific cognitive domains in veterans (+) mTBI that are similar to those with early-stage PD subjects who are almost four decades older. Age-related cognitive decline is a preclinical/prodromal, non-motor characteristic of PD. Here, we found that young veterans (+) mTBI are exhibiting some specific premature cognitive aging effects that might be considered a possible phenotype linking remote mTBI to PD in later years as reported in epidemiology studies. Data from this study provides the preliminary evidence to support a longitudinal investigation to systematically examine subtle, subclinical cognitive functioning to determine which specific domains are more likely to accurately predict risks for PD in those with mTBI.

## Acknowledgments

We wish to thank Dr. Munro Cullum for his encouragement to pursue this line of research and his editorial comments on the original manuscript. We also thank Drs. Meharvan Singh and Sid O'Bryant who provided infrastructure, intramural funding resources, and scientific support of this research. Finally, we thank the Paulie Ayala Foundation, Paulie and Leti Ayala of Punching Out Parkinson's (POPs), local POPs liaisons Don Wells, Dan Novak, and Sandy Hampton, R.N. former President of the Parkinson's Support Group of Tarrant County, and Veterans Coalition of Tarrant County who supported our community-based participatory recruitment efforts.

## Author Contributions

**Conceptualization:** Vicki A. Nejtek.

**Data curation:** Vicki A. Nejtek, Rachael N. James, Helene M. Alphonso.

**Formal analysis:** Vicki A. Nejtek, Rachael N. James.

**Funding acquisition:** Vicki A. Nejtek.

**Investigation:** Vicki A. Nejtek, Helene M. Alphonso.

**Methodology:** Vicki A. Nejtek.

**Project administration:** Vicki A. Nejtek, Rachael N. James.

**Supervision:** Vicki A. Nejtek.

**Writing – original draft:** Vicki A. Nejtek, Michael F. Salvatore.

**Writing – review & editing:** Vicki A. Nejtek, Rachael N. James, Michael F. Salvatore, Gary W. Boehm.

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
