## [Decision Letter · Decision Letter 0]

11 May 2021

PONE-D-21-11322

Premature Cognitive Aging in Specific Domains in Young Veterans with mTBI

Coincide with Early-Stage Parkinson’s Disease

PLOS ONE

Dear Dr. Nejtek,

Thank you for submitting your manuscript to PLOS ONE. After careful consideration, we feel that it has merit but does not fully meet PLOS ONE’s publication criteria as it currently stands. Therefore, we invite you to submit a revised version of the manuscript that addresses the points raised during the review process.

We look forward to receiving your revised manuscript.

Kind regards,

Firas H Kobeissy, PhD

Academic Editor

PLOS ONE

Journal Requirements:

3. Please include your tables as part of your main manuscript and remove the individual files. Please note that supplementary tables (should remain/ be uploaded) as separate "supporting information" files.

Reviewers' comments:

Reviewer's Responses to Questions

**Comments to the Author**

1. Is the manuscript technically sound, and do the data support the conclusions?

Reviewer #1: Partly

Reviewer #2: Partly

2. Has the statistical analysis been performed appropriately and rigorously? 

Reviewer #1: Yes

Reviewer #2: No

3. Have the authors made all data underlying the findings in their manuscript fully available?

Reviewer #1: Yes

Reviewer #2: Yes

4. Is the manuscript presented in an intelligible fashion and written in standard English?

Reviewer #1: Yes

Reviewer #2: Yes

5. Review Comments to the Author

Reviewer #1: This manuscript reports the results of a study comparing cognitive performance of veterans with and without mTBI, patients with early-stage Parkinson’s disease, and healthy controls. The authors report subtle cognitive deficits in mTBI+ with performance levels observed that were similar to PD and conclude this is evidence of premature cognitive aging in veterans with mTBI.

This study contributes to the literature exploring the long-term cognitive consequences of mTBI. While the manuscript is generally well-written, there are some points of clarification or expansion that would improve the interpretability of the reported results. These are described below.

1) Minor suggestion – It might be clearer and simpler for the reader if the terms mTBI+ and mTBI- were used throughout rather than ‘veterans (+) mTBI’ and ‘veterans (-) mTBI’.

2) Additional description/characterization of the mTBI+ group is needed. At a minimum, the authors should report results of the mTBI questionnaires that were administered so that the reader can better understand the mTBI sample.

3) Please be more clear about the specific comparisons of mTBI+ with both PD and mTBI-. Focus is placed on comparisons to healthy controls and PD; however, there appear to be some interesting findings regarding lack of differences from mTBI-. Further, mTBI- also appears to differ from controls on some of the tests. For example, CLOX2 scores for mTBI-, mTBI+, and PD are all lower than controls but similar to each other. Also, animal naming scores in mTBI- are lower than controls and similar to mTBI+ and PD. However, little attention is paid to this lack of difference in the reporting of the results. Further, possible explanations for this should be elaborated upon in the Discussion. For example, what might it mean that mTBI+ and PD performance is lower compared to controls but not different from mTBI-? How are we to believe that the mTBI is the issue in particular cognitive domains when performance does not differ between mTBI+ and mTBI- on some tests?

4) If available, the authors should provide some comparative data for other early-stage PD samples on the selection of cognitive tests administered. This information would help the reader place the performance of this particular sample in context. Is this sample typical of other early-stage PD samples in their performance on these tests? Similarly, is there additional data for this selection of cognitive tests in comparable mTBI+ samples?

5) Figures for animal naming and CLOX2 scores should be added. As it, the figures seem to be highlighting those tests where there were differences from mTBI-, but it is equally important for the reader to see the patterns of performance between the groups on the other tests.

6) A table showing the frequencies of low performance (using standard performance thresholds – e.g., <10th percentile, <2nd percentile, etc) in each group would enhance the ability to interpret the findings regarding performance relative to norms.

Reviewer #2: This study examined the cognitive performance among four groups (Veterans +/- mTBI, healthy controls and PD) in order to “identify specific links between mTBI and PD” and “to characterize the earliest cognitive manifestations of mTBI-related PD”. The study design is novel and addresses a gap in the literature. The article is clear and well written; however, there are several methodological and interpretative concerns, which are enumerated below.

Methods

1. On p. 3, the study is reported to be “An exploratory, matched-control, cross-sectional study matching for age”, yet, upon further reading, the PD group was not matched to the other 3 groups by age. Please reconcile this apparent discrepancy.

2. Were groups matched on Full as well as Verbal and Performance IQ? Given EF and visual spatial functions can be impaired in PD, Verbal IQ (or education level) may be a more suitable variable.

3. Were the PD participants Veterans? This wasn’t clear. Veterans may represent a unique population, which could limit the interpretation of the comparisons between non-Veteran and Veteran groups.

4. What is the rationale for selecting PD participants without a history of mTBI? If the goal of the study is to “characterize the earliest cognitive manifestations of mTBI-related PD”, wouldn’t a PD group with a history of mTBI be more appropriate?

5. Please note the MMSE cut off used to determine dementia for all groups on (on p. 4).

6. Did mTBI Veterans need to meet all 4 criteria listed on p. 5 for diagnosis? If not, rephrasing of the criteria to include “and/or” is suggested. GCS scores are extremely difficult to obtain in this population. If GCS scores were used, including the number of Veterans diagnosed using GCS criteria would be of interest.

7. Continuous variables are typically preferred over categorical when available. Why did the authors choose to categorize the PCL and GDS rather than keeping them continuous in the analyses? Also, are these categorical splits validated in both TBI and PD?

8. Given that motor symptoms could impact some of the cognitive tests such as TMT, why was motor function not controlled for in the analyses?

9. The Methods described converting GDS and PCL to categorical variables, but on p. 6 of the statistical analysis section, it is stated that a pearson correlation was used to compare them.

10. Given that the groups differed on mood and age, what is the rationale for not conducting a MANCOVA controlling for these variables?

Results

11. Did any group(s) contain more outliers than the other group(s)?

12. The demographic and clinical characteristics provided for the groups (p. 7-8) are sparse. As noted on p. 5, the authors conducted a thorough TBI evaluation with several measures (e.g., BAT-L), yet information on the TBIs (e.g., injury type, number of TBIs, time since most recent/significant TBI) is not provided. Likewise, there is minimal information on the PD group. What is the duration of PD diagnosis? Are UPDRS-Part 3 scores available? MMSE scores were provided for the PD group, but what were the MMSE scores for the other 3 groups? Were MMSE scores significantly different among the 4 groups? What were the average PCL and GDS scores per group? A table providing this information may be helpful.

13. On p. 8, it is stated that, “There was no influence of medication on outcomes (data not shown).” Were these correlations? It is recommended to include the lowest p value (i.e., “all p’s > .xx) in lieu of the statement “data not shown” for full transparency.

14. Did any other groups besides +mTBI significantly differ from each other in terms of mood?

15. Were the group x PCL and group x GDS interactions significant for the individual cognitive tests? What is the rationale for examining the interactions of the mood variables using “the combined cognitive tests” when this composite was not examined in the main analyses? Why was PCL included as a covariate in the GDS model? This doesn’t seem to make sense due the high degree of multicollinearity as noted by the authors.

16. Given that mood/PTSD tends to be significantly elevated in mTBI Veterans compared to Veteran controls (and healthy controls), I wonder if these differences could be driven by mood. Did Veterans +/- TBI differ on cognitive performance when controlling for mood?

17. On p. 10, it is stated, “these data provide several indications of subclinical, yet premature cognitive aging occurring in young veterans (+) mTBI”. While this may be true, it may also indicate residual post-concussive symptoms (PCS). Tempering of this conclusion is suggested.

Discussion

18. Likewise, tempering the statement on p. 10, “These specific tests may be particularly sensitive in detecting subtle mTBI-related cognitive problems that could be used to benchmark an individual’s performance as a premature cognitive aging phenotype with a potential trajectory of PD” or providing alterative explanations is recommended. Could results also indicate residual PCS? Could they also serve as a harbinger for other neurodegenerative processes, such as DLB, AD, etc.?

19. While intriguing, the conclusions are mainly based on null findings between mTBI and the PD groups, both of which have small sample sizes and analyses that could be possibly underpowered.

20. On p. 11, it is written, “Unfortunately, the typical cognitive test used in routine healthcare settings (e.g. Mini Mental State Exam) lacks the sensitivity to identify subclinical cognitive decline.”. Given that the authors have the MMSE data on these subjects, it appears this would be a good addition to this study to confirm that this is indeed the case with their sample as well.

Figures/Tables

21. On p. 8, it is noted that Figure 1 “illustrates statistical differences”, but no statistical differences are denoted within this figure. Please include asterisks between bars that statistically differ.

22. Please consider including p values and/or effect sizes in Table 1.

6. PLOS authors have the option to publish the peer review history of their article (what does this mean?). If published, this will include your full peer review and any attached files.

Reviewer #1: No

Reviewer #2: No

---

## [Author Response · Author response to Decision Letter 0]

25 Jun 2021

PONE-D-21-11322

Premature Cognitive Aging in Specific Domains in Young Veterans with mTBI

Coincide with Early-Stage Parkinson’s Disease

PLOS ONE

Dear Dr. Kobeissy,

Thank you for allowing us to resubmit our revised manuscript to PLOS ONE. 

We have carefully and substantially revised this manuscript, figure, and table as per the Editorial style requirements and the reviewer’s concerns. We sincerely trust that we have satisfactorily addressed and appropriately answered each and every suggestion or comment made by the reviewers. 

We have included the following items for your review: 

• A rebuttal letter that responds to each point raised by the academic editor and reviewers that we labeled 'Response to Reviewers'.

• A marked-up copy of our manuscript that highlights changes we made to the original version and labeled 'Revised Manuscript with Track Changes'.

• An unmarked version of our revised paper that is clean and without tracked changes labeled as 'Manuscript'.

Below, please see our responses concerning journal requirements and to each reviewer’s comments.

We look forward to receiving your comments about our revised manuscript.

Kind regards,

Vicki A. Nejtek, PhD

Associate Professor

Journal Requirements:

 and

Authors response: Thank you for your patience with informing us about the formatting unique to PlOSOne specifications. We have reviewed these requirements and are hopeful that our revisions are now in the acceptable journal formatting style including superscript references revised to square brackets [1,2,3], headings, subheadings, line numbering, etc.

Authors response: After reviewing the title requirement, our revisions now include the title page with the manuscript itself. 

3. Please include your tables as part of your main manuscript and remove the individual files. Please note that supplementary tables (should remain/ be uploaded) as separate "supporting information" files.

Authors response: After reviewing the table embedding requirement, our revisions now include the table within the manuscript itself. 

Authors response: We have deleted the ‘data not shown’ text and have provided the data within the manuscript rather than supporting files. 

Authors response: We have deleted the ‘data not shown’ text and have provided the data within the manuscript and have no supporting files to add to this submission.

Reviewers' comments:

Reviewer's Responses to Questions

5. Review Comments to the Author

Authors Response: We have substantially revised the manuscript, figure, table as per the reviewer’s comments, and added several references. Please note that our responses here below use the line numbering in the revised highlighted manuscript (rather than the clean version) to show the reviewer’s exactly what we deleted, revised, or added.

Reviewer #1: This manuscript reports the results of a study comparing cognitive performance of veterans with and without mTBI, patients with early-stage Parkinson’s disease, and healthy controls. The authors report subtle cognitive deficits in mTBI+ with performance levels observed that were similar to PD and conclude this is evidence of premature cognitive aging in veterans with mTBI.

This study contributes to the literature exploring the long-term cognitive consequences of mTBI. While the manuscript is generally well-written, there are some points of clarification or expansion that would improve the interpretability of the reported results. These are described below.

1) Minor suggestion – It might be clearer and simpler for the reader if the terms mTBI+ and mTBI- were used throughout rather than ‘veterans (+) mTBI’ and ‘veterans (-) mTBI’.

Authors Response: We appreciate the suggestion and while we believe that the veteran designation is important to describe the subject sample, we deleted the ‘(-) mTBI’ from the veteran (-) mTBI group. Those subjects are now only referred to as ‘Veterans’ in order to help distinguish this group’s performance from non-military Healthy Controls. The veterans with mTBI remain Veterans (+) mTBI to distinguish this group from civilians.

2) Additional description/characterization of the mTBI+ group is needed. At a minimum, the authors should report results of the mTBI questionnaires that were administered so that the reader can better understand the mTBI sample.

Authors Response: Although, the mTBI questionnaires were primarily utilized to screen out more severe brain injuries than mild, we understand the reviewer’s concern that the readers need to better understand the characteristics of the sample. Thus, we have added substantial text explaining that the BAT-L was used to grade the mildness of the concussion on which we report the outcome. We also added text detailing that all the combined mTBI tools together allowed us to collect current physiological symptoms, past symptoms aligning with concussion grade, and the nature or origin of the injury (p. 7). Additionally, we begin reporting the results of these questionnaires (p.10) listing blasts/explosions, motorized accidents, physical assaults, falls as well as the mean number of mTBIs experienced during their mean years of military service, and included military branch served and areas of deployment. We added Grade I, II, or III and the list of current symptoms (line 259-265). 

3) Please be more clear about the specific comparisons of mTBI+ with both PD and mTBI-. Focus is placed on comparisons to healthy controls and PD; however, there appear to be some interesting findings regarding lack of differences from mTBI-. Further, mTBI- also appears to differ from controls on some of the tests. For example, CLOX2 scores for mTBI-, mTBI+, and PD are all lower than controls but similar to each other. Also, animal naming scores in mTBI- are lower than controls and similar to mTBI+ and PD. However, little attention is paid to this lack of difference in the reporting of the results. Further, possible explanations for this should be elaborated upon in the Discussion. For example, what might it mean that mTBI+ and PD performance is lower compared to controls but not different from mTBI-? How are we to believe that the mTBI is the issue in particular cognitive domains when performance does not differ between mTBI+ and mTBI- on some tests?

Authors Response: The authors very much appreciate the reviewer’s suggestion to be more descriptive and clearer about the performance each of the groups. The Cognitive Outcomes section (pp 12-16) was completely revised to provide much more detail about our findings. We also substantially revised the Normative Score Interpretation section (p.14-16) to help in this regard. We apologize for assuming that the reader tacitly understood that the non-mTBI veterans were also considered a control group along with the non-veteran healthy controls. On p. 5 (line 125), we added a sentence stating that veterans and non-veteran healthy controls served as controls to the veterans (+) mTBI. On p 13-14, we itemized the results and added a brief interpretation for each item to make it a bit easier for the reader to discern the importance or unimportance concerning the group differences in cognitive test performance. 

Prior to revisions, in our original text, we were unable to find the reviewer’s comment about our focus only on comparisons between healthy controls and PD. However, in reviewing this section, we realized that we had not included statistics from a 5th cognitive test primarily because the post-hoc results indicated that the only 2 groups driving the significance was between PD and healthy controls -- which is intuitive and not very informative (Item #4, p.13). Thus, in our effort to keep every results interesting and transparent, we added these results and used bulleted numbers with each result’s data interpretation for all of the outcomes to help make our findings more detailed and clearer. In the discussion section, p17 (line 411-412), we also added a paragraph revealing that on some cognitive tests all groups performed within normal limits and whatever differences we observed were unique to particular domains of cognitive functioning. 

Given the fact that we completely and substantially revised pp12-19), revised and added more detail to Fig 1 and Table 1 and also augmented our previous data interpretations in our revision, we hope that we have now answered the critiques sufficiently to satisfy the reviewer. 

4) If available, the authors should provide some comparative data for other early-stage PD samples on the selection of cognitive tests administered. This information would help the reader place the performance of this particular sample in context. Is this sample typical of other early-stage PD samples in their performance on these tests? Similarly, is there additional data for this selection of cognitive tests in comparable mTBI+ samples?

Authors Response: Unfortunately, we could not find studies examining cognitive testing specifically in early-stage PD… other more advanced stages, yes, but not early-stage. If compared to more advanced stages, then we believe it would add confusion to our study goals and data interpretations. For use in mTBI, we state on p.7, “Standardized cognitive assessments used in this study (1) are listed in the National Institute of Neurological Disorders and Stroke Common Data Elements, (2) have been extensively used in neurodegenerative disease and aging research, and (3) have well-known reliability and validity coefficients.” Also, Writer et al50 reported on CLOX performance in mostly mTBI in terms of real-world competency which we mention in the discussion, pp.19-20 (line 483-493). 

However, the revision of Table 1 shows the available percentiles stratified by age and education for 4 cognitive tests that we used for the all groups. We were forced to use normative mean scores for the other 4 tests that we could find that were unique to the age and education level of the PD groups. All references used to construct Table 1 are shown within the table as superscripts. We believe with the extensive revisions we created on pp 11-19, have probably addressed this comment to the best of our ability and hope that our revisions have helped satisfy the reviewer. Also, p. 18 (line 434-446) interprets the cognitive findings for PD subjects and on p.20 (491) in the Discussion states, “Similarly, in a PD population, these same cognitive skills are necessary to make strategic vs. careless decisions influencing therapeutic compliance vs. non-compliance impacting disease progression”

5) Figures for animal naming and CLOX2 scores should be added. As it, the figures seem to be highlighting those tests where there were differences from mTBI-, but it is equally important for the reader to see the patterns of performance between the groups on the other tests.

Authors Response: We appreciate the reviewer’s suggestion and considered adding figures for the other tests. However, we planned to highlight these tests since TMT-A and TMT-B are the primary outcomes, and we therefore hesitated in creating more figures for secondary outcomes that were also displayed in great detail in the revised Table 1. We are therefore hopeful that the Table 1 revision, and the substantial revised text pp 11-19 is satisfactory for the reviewer. 

6) A table showing the frequencies of low performance (using standard performance thresholds – e.g., <10th percentile, <2nd percentile, etc) in each group would enhance the ability to interpret the findings regarding performance relative to norms.

Authors Response: We appreciate the reviewer’s expertise in neuropsychology in suggesting we make the normative scores more meaningful in relation to each groups mean score. To simplify the readability of Table 1, we have substantially revised this table so that readers can see at a glance the exact percentile corresponding to each group’s raw mean score. On pp.14-16 we substantially added text concerning the Normative score interpretation to help elucidate our findings. We hope this additional information along with an easier to digest Table helps assuage the reviewer’s concern and addresses their excellent suggestion.

Reviewer #2: This study examined the cognitive performance among four groups (Veterans +/- mTBI, healthy controls and PD) in order to “identify specific links between mTBI and PD” and “to characterize the earliest cognitive manifestations of mTBI-related PD”. The study design is novel and addresses a gap in the literature. The article is clear and well written; however, there are several methodological and interpretative concerns, which are enumerated below.

Methods

1. On p. 3, the study is reported to be “An exploratory, matched-control, cross-sectional study matching for age”, yet, upon further reading, the PD group was not matched to the other 3 groups by age. Please reconcile this apparent discrepancy.

Author Response: We have revised the order of the paragraph and language to bring better attention to the goal of the study which was to compare cognitive outcomes in young veterans + mTBI… to older early-stage PD subjects who were IQ-matched to the other groups (p.4, line 95). This goal was also mentioned in the Abstract in the ‘Participants’ section (p. 2 line 39), and discussed in the introduction last paragraph (p. 4) as a roadmap to guide the reader towards understanding the importance of determining the presence of premature cognitive decline that would not have been possible without a comparison to an older PD group. 

 2. Were groups matched on Full as well as Verbal and Performance IQ? Given EF and visual spatial functions can be impaired in PD, Verbal IQ (or education level) may be a more suitable variable.

Authors Response: Estimating IQ with the AMNART requires that pronunciation errors and education years are used in a formula to render verbal IQ. As the Reviewer points out, verbal IQ and education level may be more suitable variables. Fortunately, we used the AMNART (90 point minimum) verbal IQ and education (12-years minimum) for the inclusion/exclusion criteria prior to study enrollment (p. 5, line 121-124). As this is a well-known and commonly used tool in aging and Alzheimer’s research, we sought to remain consistent with the existing research in these fields. 

3. Were the PD participants Veterans? This wasn’t clear. Veterans may represent a unique population, which could limit the interpretation of the comparisons between non-Veteran and Veteran groups.

4. What is the rationale for selecting PD participants without a history of mTBI? If the goal of the study is to “characterize the earliest cognitive manifestations of mTBI-related PD”, wouldn’t a PD group with a history of mTBI be more appropriate?

Authors Response to #3 and #4: Indeed, the reviewer accurately points out and we agree that veterans may represent a unique population. Thus, we planned to compare them to age- and IQ-matched veterans without mTBI as well as non-veteran healthy controls to help account for any unique veteran-related influence. 

We appreciate the reviewer’s vision of using a mTBI-related PD group. However, this is a preliminary study and the first of its kind to compare older early-stage PD subjects to younger veterans + mTBI. Considering our goal was to identify premature cognitive decline as an advanced aging (yet subtle) artifact already present (albeit not clinically recognized) in young veterans + mTBI, we did not seek to add an additional neurological insult to an existing neurologically compromised disease group for comparison. We believe a substantial increase to our sample size would be required and this would no longer be a preliminary or hypothesis-generating study – this would also change our primary goal. 

If we had allowed mTBI in the PD group (to maintain the integrity of the matched study design) we would have needed to recruit age-and IQ-matched veterans with early-stage PD with a recent history of mTBI to match our current veteran sample. Also, as PD subjects that we sought to recruit were older, the period of time in the military would have been decades earlier, and the length of time elapsed from the original neural insult would most likely have also been several decades earlier. We believe that having decades older mTBI would be difficult to assess, verify, and would more likely address a much more residual neural effect that is beyond the scope of this investigation. Further, at this stage in their lives, the older PD group with a diagnosis of mTBI (also burdened with a PD diagnosis) would have been difficult (if not impossible) for us to determine which pathology was more influential –the mTBI, age-related decline, or PD. Thus, parsing out the contribution of mTBI, PD pathology and age would have also changed the study goals and design.

 Nevertheless, in our original text, we had stated, “Veterans (+) mTBI were exposed within the past 7-years during their military career to non-penetrating mTBI. Healthy controls and PD subjects had no recent military service or mTBI histories” (p.6 line 144-146)). For clarity, we have also added the text – ‘older’ to the PD group (p.3,4, line 74,95,98,106). We hope these revisions and explanations help address the reviewer’s insightful suggestions. 

5. Please note the MMSE cut off used to determine dementia for all groups on (on p. 4).

Author’s response: We added the minimum score of 24 (p.6, line 140) as per Crum et al (1993) and Folstein et al (1975). We also highlighted the text indicating further exclusion of dementia (p.5,line 130) and mild cognitive impairment (p.6, line 140,141).

6. Did mTBI Veterans need to meet all 4 criteria listed on p. 5 for diagnosis? If not, rephrasing of the criteria to include “and/or” is suggested. GCS scores are extremely difficult to obtain in this population. If GCS scores were used, including the number of Veterans diagnosed using GCS criteria would be of interest.

Author’s response: We appreciate the reviewer’s suggestion and have revised the text (p.5) to reflect “and/or”. We stated that the GCS could be used (if available) (p.7, line 165) and that these criteria were used to screen out those with more severe head trauma than mTBI…‘prior to study enrollment’ (p.7, line 161). 

7. Continuous variables are typically preferred over categorical when available. Why did the authors choose to categorize the PCL and GDS rather than keeping them continuous in the analyses? Also, are these categorical splits validated in both TBI and PD?

Author’s response: We appreciate the reviewer asking these important questions. In designing the study, we were cognizant that both veteran groups and especially veterans + mTBI may report more PTSD symptoms than the other subject groups. We were (are) also aware that depression is inherently embedded within a PTSD diagnosis. Thus, our first concern was to limit the risk of enrolling subjects where PTSD or depression symptoms would be clinically significant enough to interfere with cognitive performance. 

Secondly, we referred to the National Center for PTSD accessed from www.ptsd.va.gov and Weather et al., 1993 that state that all 17-items can be summed as raw total scores or as a cluster of psychophysiological symptoms using the number of symptoms within each cluster at the moderate or greater level. This makes a categorical level of data interpretation more informative to identify who endorses clinically significant PTSD symptoms that would qualify for a diagnosis versus transient symptoms due to dynamic and common life stresses. 

 Third, as the PCL has a minimum raw score of 17 points/item indicating “not at all” as having any PTSD symptoms, it is quite possible that using continuous raw scores could artificially inflate the statistical parameters especially when entered into a multivariate model or post hoc comparisons that might lead to a violation of our assumptions that there was an effect of these 17 points when actually there were little or no symptoms as indicated in our raw scores for healthy controls (M=20.5) and PD subjects (M=19.3). 

Although the PCL can be a summed total score, the manner in which this total score is categorized into severity levels is not well articulated in the literature. In revising this section (pp.7-8) we found that what was reported in the text was not only typographically incorrect, but the score range was reported by a commercial clinical treatment center rather than from an empirical source. The revised text is now corrected following the guidelines from the National Center for PTSD accessed from www.ptsd.va.gov along with Weathers et al (1993) to describe the severity level for categorical splits for data analyses and cut scores indicating clinical criteria for a PTSD diagnosis as per DSM-IV criteria. PCL cut scores for a PTSD diagnosis are 44 for the general public and 50 for military personnel (p.8, line 188, 189).

Due to the reviewer’s concerns/questions, we kept the grouping variable for data analyses, but added text to better describe the sample (p. 8): “Out of a possible high of 85, the mean score for veterans (+) mTBI was (M=38.2) and veterans (-) mTBI was (M=30.7). These mean scores are lower than the cut scores required for a PTSD diagnosis. 

For depression, the text we currently have for the GDS (p.8, line 190-193) is categorized into severity levels as per Yesavage et al., 1983. After considering the reviewer’s insightful comments we deleted the existing text and better clarified the findings reported in the results section 2nd paragraph in the Posttraumatic Stress and Depression section (p.8). The corrected revision provides the mean raw score for each group which more accurately describes the sample as within the normal range well as reporting the GDS results when PCL levels are treated as a covariate. 

Finally, while the categorical split for PCL has been used in veterans and mTBI, we are unaware of its use in PD subjects. Similarly, we are unaware of GDS use in young veterans with or without mTBI, or its use with young civilian healthy controls as this instrument is normally used in geriatric populations. This is stated on p. 11 (line 276-277) under the Posttraumatic Stress and Depression results section.

8. Given that motor symptoms could impact some of the cognitive tests such as TMT, why was motor function not controlled for in the analyses?

Author’s response: Our exclusion criteria listed on p. 4 states…” had any physical or sensory limitations preventing completion of study procedures”. While we were interested in early-stage PD, we were aware that we needed to limit enrollment to only those PD subjects who were in the earliest stages of the disease who had the physical and sensory capabilities to perform all of the cognitive assessments to the best of their abilities without additional interference from motor impairment. Additionally, the UPDRS was used as a screening instrument for our study physician along with our consensus judgement. However, we added text elucidating that the scores on the UPDRS indicated subjects had little or no motor impairments and were not worse than early-stage (p.6, line 150-152), and we added text in the limitation section (p. 21, line 513-521). Thus, there was no need to statistically control for motor function’s influence on cognition.

9. The Methods described converting GDS and PCL to categorical variables, but on p. 6 of the statistical analysis section, it is stated that a pearson correlation was used to compare them.

Author’s response: We distinguish between raw scores and severity levels (p. 9, line 214-218) which now reports the raw score group means for the PCL as well as severity levels and cut scores (p. 11, line 279-280) to inform the reader about the scores necessary to classify subjects as having a PTSD diagnosis and how the statistics were used. The GDS also now states raw scores which were not ‘converted’, but categorized into normal, mild, or severe as instructed in the literature. The Pearson correlation was conducted using raw scores (p.11-12, line 294, 295).

10. Given that the groups differed on mood and age, what is the rationale for not conducting a MANCOVA controlling for these variables?

Author’s response: To better clarify the issue with mood (we presume the author is referring to PTSD), we retrieved the raw scores for each subject and found additional information from www.ptsd.va.gov concerning the cut scores to meet a PTSD diagnosis (44 for general population and 50 for veterans). According to these cut scores, our subjects did not meet diagnostic criteria for PTSD. Additionally, our Exclusion criteria states that…”psychiatric or neurological diseases / disorders (other than mTBI or early stage PD)” are not allowed study entry (p.4, line 128-129). And on p. 5, we state that the MINI was used to screen out…”current and unremitted mood and substance…disorders…” The PCL levels of severity were also treated as an ordinal variable that could be analyzed as a covariate to examine its potential to influence cognitive outcomes in multivariate analyses – [aka MANCOVA] as stated on p.9 (line 217-218).

As for age, as addressed earlier, we always intended for a young to old comparison to help identify if young veterans (+) mTBI were performing cognitive tests as if they were decades older and had PD. Please also refer to our previous response to Item #1. 

Results

11. Did any group(s) contain more outliers than the other group(s)?

Author’s response: On p. 7, we state…“Out of 123 participants, there were 9 excluded outliers whose cognitive test scores were > 2 standard deviations above or below the standardized normative test values.” We added, (2 HC, 1 veteran (-) mTBI, 3 veterans (+) mTBI and 3 subjects in the PD group) on p.9, line 234. 

12. The demographic and clinical characteristics provided for the groups (p. 7-8) are sparse. As noted on p. 5, the authors conducted a thorough TBI evaluation with several measures (e.g., BAT-L), yet information on the TBIs (e.g., injury type, number of TBIs, time since most recent/significant TBI) is not provided. Likewise, there is minimal information on the PD group. What is the duration of PD diagnosis? Are UPDRS-Part 3 scores available? MMSE scores were provided for the PD group, but what were the MMSE scores for the other 3 groups? Were MMSE scores significantly different among the 4 groups? What were the average PCL and GDS scores per group? A table providing this information may be helpful.

Author’s response: On pp.10,11 (line 252-265), we have added more detail about the veterans, mTBI, and on p. 6 (line 150-159) we added more information about how the UPDRS was used in the PD group. The years since the most recent mTBI is 7-years or less as stated in our inclusion/exclusion criteria, p.6, line 143-146. We added text about number of mTBI, injury type, military branch, years served, etc. p. 10-11, line (252-265) The PCL and GDS raw scores have been added (pp. 9, 10).

Mean MMSE scores were used as a screening tool to remain consistent with the aging, Alzheimer’s and PD research literature. In contrast to MMSE (a rather insensitive global measure of cognitive functioning), we were interested in examining specific domains of cognitive functioning that would typically be overlooked in a geriatric primary care clinic. As the MMSE is the standard cognitive test given to geriatric patients, younger than middle age veterans would probably never receive such an assessment. As indicated in some of the recent literature and as we suspected, the MMSE is insensitive to specific cognitive domains and we found that it missed important limitations in cognitive functioning that more specific cognitive tests detected (p.21, line 525-526), p. 14, line 317-343). Moreover, the purpose of the MMSE was to determine if subjects were within the normal range of general, global cognitive functioning prior to enrollment, and to exclude subjects who had MCI. The MMSE results show that no one had MCI or dementia by its standard cut scores, p. 10 line 239-240. Taken together, this instrument was never intended to be statistically analyzed beyond that as a global screening instrument typically used in the clinic setting. 

13. On p. 8, it is stated that, “There was no influence of medication on outcomes (data not shown).” Were these correlations? It is recommended to include the lowest p value (i.e., “all p’s > .xx) in lieu of the statement “data not shown” for full transparency.

Author’s response: On p.11, line 266-272, we list the statistics showing no influence of medication on cognitive outcomes: There was no influence of medication on TMT-A [F(1,25)=.01, p=.92], TMT-B [F(1,25)=.12, p= .73], CLOX1 [F(1,25)=3.26, p=.083], CLOX2 [F(1,25)=.027, p=.87], WMS III number of correct total FWD & BWD [F(1,25)=2.35, p=.132], FAS [F(1,25)=.004, p=.95], or Animal Naming [F(1,25)=.09, p=.77]. 

14. Did any other groups besides +mTBI significantly differ from each other in terms of mood?

Author’s response: Please see our previous response to question 10 related to mood and our extensive revised text on pp. 4-7, 9,10.

15. Were the group x PCL and group x GDS interactions significant for the individual cognitive tests? What is the rationale for examining the interactions of the mood variables using “the combined cognitive tests” when this composite was not examined in the main analyses? Why was PCL included as a covariate in the GDS model? This doesn’t seem to make sense due the high degree of multicollinearity as noted by the authors.

Author’s response: On p.11, 12, line 275-299, we stated that there was no significant interaction of PCL levels x Group (not GDS) on cognitive tests. We appreciate the reviewer pointing out the poor use of terminology – “combined” and have revised the sentence as follows: “Further, we found no interaction effect of PCL levels on Group performance across the different cognitive tests multivariate analyses examining PCL severity levels x Group influence on the combined cognitive tests revealed no significant interaction [F (21, 285) =.897, p=.596; Wilks' � = .832, partial �2 =.059].” Also, we appreciate the reviewer’s comment on the PCL covariate stats. We have deleted the statistical covariate sentence and also revised the last sentence as follows: Based on the above results illustrating all groups had GDS scores within the normal range and the high degree of multicollinearity found between PCL and GDS, no further analyses were conducted using GDS or PCL scores.

16. Given that mood/PTSD tends to be significantly elevated in mTBI Veterans compared to Veteran controls (and healthy controls), I wonder if these differences could be driven by mood. Did Veterans +/- TBI differ on cognitive performance when controlling for mood?

Author’s response: Please see our extensive answers to mood and PTSD outcomes for questions 10 and 14, and the substantial changes we made in the paper pp 9-10, line 234-237. 9911-12, line 276-294.

17. On p. 10, it is stated, “these data provide several indications of subclinical, yet premature cognitive aging occurring in young veterans (+) mTBI”. While this may be true, it may also indicate residual post-concussive symptoms (PCS). Tempering of this conclusion is suggested.

Author’s response: Throughout the paper we have revised the conclusion for Parkinson’s and have included a general statement about the possibility of “other neurodegenerative process” 

Discussion

18. Likewise, tempering the statement on p. 10, “These specific tests may be particularly sensitive in detecting subtle mTBI-related cognitive problems that could be used to benchmark an individual’s performance as a premature cognitive aging phenotype with a potential trajectory of PD” or providing alterative explanations is recommended. Could results also indicate residual PCS? Could they also serve as a harbinger for other neurodegenerative processes, such as DLB, AD, etc.?

Author’s response: On p. 2; p.18, p. 21, we have revised the text to include neurodegeneration, dementia, PD or some other disease process. We hesitated to add specific disease processes like DLB which we cannot assess since dementia and MCI were excluded. Thus, we chose to be more inclusive of broader pathologies. 

19. While intriguing, the conclusions are mainly based on null findings between mTBI and the PD groups, both of which have small sample sizes and analyses that could be possibly underpowered.

Author’s response: On p. 21 in the Limitations and Strength section, the first sentence states that in this exploratory study our small sample size is a limitation. We have substantially added text to further elucidate the findings and the potential it has to inform the science while acknowledging the limitations of an exploratory study. Actually, throughout the results, we list the premature aging effects that can be gleaned by studying Table 1 and the normative data as well as the differences between the veterans (+) mTBI and their respective controls groups rather than relying solely on the PD comparison. We hope our revisions have satisfied the reviewer as we incorporated most all of their excellent suggestions. 

20. On p. 11, it is written, “Unfortunately, the typical cognitive test used in routine healthcare settings (e.g. Mini Mental State Exam) lacks the sensitivity to identify subclinical cognitive decline.”. Given that the authors have the MMSE data on these subjects, it appears this would be a good addition to this study to confirm that this is indeed the case with their sample as well.

Author’s response: On p. 10, line 239-241 we added MMSE means. Also p.22 now contains, “Especially important was realizing that the MMSE scores, verbal phonetic fluency, working memory for digits tests failed to identify subtle cognitive problems.” Thank you for bringing this to our attention.

Figures/Tables

21. On p. 8, it is noted that Figure 1 “illustrates statistical differences”, but no statistical differences are denoted within this figure. Please include asterisks between bars that statistically differ.

Author’s response: We have added p-values to Figure 1 

22. Please consider including p values and/or effect sizes in Table 1.

Author’s response: As the Table 1 is primarily providing standardized normative data to describe how the raw scores align with the general population, we believe it could be misleading to the reader to provide significance and effect sizes for group comparison raw scores that are already available within the text Results section. Here, we provided additional information that is not in the text – Confidence Intervals and the percentiles and give a brief interpretation in the results that should help clarify our findings. We chose to show these data so that the reader would have the perspective of the standardized normative score range with percentiles to guide them towards understanding how the raw scores for each of the groups and especially the illustrates the subtleties of non-clinical cognitive decline.

6. PLOS authors have the option to publish the peer review history of their article (what does this mean?). If published, this will include your full peer review and any attached files.

Do you want your identity to be public for this peer review? For information about this choice, including consent withdrawal, please see our Privacy Policy.

Reviewer #1: No

Reviewer #2: No

---

## [Decision Letter · Decision Letter 1]

31 Aug 2021

PONE-D-21-11322R1

Premature Cognitive Aging in Specific Domains in Young Veterans with mTBI

Coincide with Early-Stage Parkinson’s Disease

PLOS ONE

Dear Dr. Nejtek,

Thank you for submitting your manuscript to PLOS ONE. After careful consideration, we feel that it has merit but does not fully meet PLOS ONE’s publication criteria as it currently stands. Therefore, we invite you to submit a revised version of the manuscript that addresses the points raised during the review process.

We look forward to receiving your revised manuscript.

Kind regards,

Firas H Kobeissy, PhD

Academic Editor

PLOS ONE

Journal Requirements:

Additional Editor Comments:

Dear Vicki,

I apologize that it took longer time for the review,

the comments are encouraging,

the third reviewer emailed back and she had problem with logging on and accessing:

the comments are valid and I think they can be rectified:

1. There is definitely no study out there that indicates that 56% of Veterans develop PD within 4.6-6.8 years of mTBI. If this were the case, we would have a massive epidemic of PD right now. In fact, PD is a relatively rare disease with incidence <<1%, even after TBI. Perhaps the authors mis-understood an epidemiological study that may have identified 56% increased RISK of PD among Veterans with mTBI.

2. The finding that in this cross sectional study, Veterans with TBI perform worse than Veterans without TBI does not equal premature "cognitive aging" as the authors have not demonstrated any progressive cognitive decline. Perhaps the Veterans with TBI simply have a static injury that is resulting in worse cognitive performance. I would not equate this with cognitive aging, which is generally a term used to indicate a progressive decline in cognition associated with aging.

Looking forward for your response

Many thanks

Reviewers' comments:

Reviewer's Responses to Questions

**Comments to the Author**

1. If the authors have adequately addressed your comments raised in a previous round of review and you feel that this manuscript is now acceptable for publication, you may indicate that here to bypass the “Comments to the Author” section, enter your conflict of interest statement in the “Confidential to Editor” section, and submit your "Accept" recommendation.

Reviewer #1: All comments have been addressed

Reviewer #2: (No Response)

2. Is the manuscript technically sound, and do the data support the conclusions?

Reviewer #1: Yes

Reviewer #2: Yes

3. Has the statistical analysis been performed appropriately and rigorously? 

Reviewer #1: Yes

Reviewer #2: Yes

4. Have the authors made all data underlying the findings in their manuscript fully available?

Reviewer #1: Yes

Reviewer #2: Yes

5. Is the manuscript presented in an intelligible fashion and written in standard English?

Reviewer #1: Yes

Reviewer #2: Yes

6. Review Comments to the Author

Reviewer #1: (No Response)

Reviewer #2: The authors addressed the majority of concerns raised and their revisions have substantially improved their manuscript. There are several remaining issues that need to be addressed:

Abstract

1. Please check this statement for accuracy: “Epidemiologists report 56% of veterans with (+) mild traumatic brain injury (mTBI) develop Parkinson’s disease (PD) within 4.6 to 6.8-years post-injury”.

Introduction

2. Please check this statement for accuracy: “Epidemiologists report that veterans (+) mTBI have a 56% increased risk of developing idiopathic Parkinson’s disease (PD) within ~4.6-years post-injury [1].” I believe that this statement may not be entirely accurate. My understanding of Gardner’s study is that the risk if increased by 56% within (up to) 12 years.

3. The new sentence, “In summary, it is uncertain whether global or specific cognitive problems lead to mTBI-related risks for PD onset” is unclear. Are the authors proposing that post-mTBI cognition is the cause for PD onset? Please revise for clarity.

Results

4. The normative table was cut-off and was unable to be reviewed.

5. Re: Response to Reviewer 1’s comment 4. There are quite a few studies that have examined cognitive functioning in early-stage PD. For example, Pfeiffer et al., 2014: Acta Neurol Scand 2014: 129: 307–318 DOI: 10.1111/ane.12189.

7. PLOS authors have the option to publish the peer review history of their article (what does this mean?). If published, this will include your full peer review and any attached files.

Reviewer #1: No

Reviewer #2: No

---

## [Author Response · Author response to Decision Letter 1]

5 Oct 2021

Additional Editor Comments:

1. There is definitely no study out there that indicates that 56% of Veterans develop PD within 4.6-6.8 years of mTBI. If this were the case, we would have a massive epidemic of PD right now. In fact, PD is a relatively rare disease with incidence <<1%, even after TBI. Perhaps the authors mis-understood an epidemiological study that may have identified 56% increased RISK of PD among Veterans with mTBI.

Author’s Response

We sincerely appreciate and thank the Editor for pointing out our unintended omission of RISK in the first sentence in the Abstract. We surely did not intend to insinuate that there was this overwhelmingly high number of Parkinson’s disease affecting our veterans. We have revised the Importance section in the Abstract lines 30-34. After careful review of our paper, we found in the Introduction, the first sentence correctly stated the 56% increased risk in lines 56-57, and we found no other miscommunications on the ‘risk’ aspect in rest of the paper. 

2. The finding that in this cross-sectional study, Veterans with TBI perform worse than Veterans without TBI does not equal premature "cognitive aging" as the authors have not demonstrated any progressive cognitive decline. Perhaps the Veterans with TBI simply have a static injury that is resulting in worse cognitive performance. I would not equate this with cognitive aging, which is generally a term used to indicate a progressive decline in cognition associated with aging.

Author’s Response

Many thanks to the Editor regarding their insight concerning the traditional use of the term ‘cognitive aging’. Although we did not run a longitudinal study and cannot predict measure progression, we used the TMT-A and B which has been investigated as a significantly sensitive predictor of progression to dementia within 4.3- to 6-years [DeRoy et al., 2020; Marchand et al, 2018, respectively] (p. 17, lines 419-429). In doing so, we might loosely extrapolate the possibility that problems with cognitive flexibility may serve as a prodromal phenotype. 

Nevertheless, we revised the title, the Conclusions and Relevance section in the Abstract (line 50) to tone down the aging aspect and revised text to state “premature cognitive decline” rather than ‘aging’. Additionally, on p. 16 (line 387), p. 17 (line 410) refers only to a future need for benchmarking cognitive functioning, and p.19 (line 478), and p. 21 (line 509), we refer to the veterans (+) mTBI performance as either having a ‘premature cognitive decline” (line 390) or ‘age-related cognitive decline’ rather than ‘cognitive aging’ per se. We also state that our results occur only in specific domains in addition to recognizing we cannot infer progression of aging. For example, p. 20, line 492-495 states, “another limitation of this study is the cross-sectional design which prevents us from longitudinally tracking the onset or progression of premature cognitive decline nor predicting PD risks in those with mTBI.”

We ask the reviewer to consider our comparisons of veterans (+) mTBI to age-matched healthy controls, veterans, decades older Parkinson’s disease subjects in addition to elder normative control data (p.13-14 and Table 1). Our comparison outcomes suggest the presence of some premature cognitive aging ‘effect’ is indeed occurring in our subjects whose mTBI occurred approximately 5.1 +/-1.9 years ago. Thank you.

Reviewers' comments:

Reviewer's Responses to Questions-

Comments to the Author

1. If the authors have adequately addressed your comments raised in a previous round of review and you feel that this manuscript is now acceptable for publication, you may indicate that here to bypass the “Comments to the Author” section, enter your conflict of interest statement in the “Confidential to Editor” section, and submit your "Accept" recommendation.

Reviewer #1: All comments have been addressed

Reviewer #2: (No Response)

2. Is the manuscript technically sound, and do the data support the conclusions?

Reviewer #1: Yes

Reviewer #2: Yes

3. Has the statistical analysis been performed appropriately and rigorously?

Reviewer #1: Yes

Reviewer #2: Yes

4. Have the authors made all data underlying the findings in their manuscript fully available?

Reviewer #1: Yes

Reviewer #2: Yes

5. Is the manuscript presented in an intelligible fashion and written in standard English?

Reviewer #1: Yes

Reviewer #2: Yes

6. Review Comments to the Author

Reviewer #1: (No Response)

Reviewer #2: The authors addressed the majority of concerns raised and their revisions have substantially improved their manuscript. There are several remaining issues that need to be addressed:

Abstract

1. Please check this statement for accuracy: “Epidemiologists report 56% of veterans with (+) mild traumatic brain injury (mTBI) develop Parkinson’s disease (PD) within 4.6 to 6.8-years post-injury”.

Author’s Response

We sincerely appreciate and thank the reviewer pointing out the serious omission of RISK we accidentally missed in the first sentence in the Abstract. We surely did not intend to insinuate that there was this overwhelmingly high number of Parkinson’s disease affecting our veterans. We have revised the Importance section in the Abstract lines 30-34. After careful review of our paper, we found in the Introduction, the first sentence correctly stated the 56% increased risk in lines 56-57, and we found no other miscommunications on the ‘risk’ aspect in rest of the paper. 

Introduction

2. Please check this statement for accuracy: “Epidemiologists report that veterans (+) mTBI have a 56% increased risk of developing idiopathic Parkinson’s disease (PD) within ~4.6-years post-injury [1].” I believe that this statement may not be entirely accurate. My understanding of Gardner’s study is that the risk if increased by 56% within (up to) 12 years.

Author’s Response

Gardner et al., 2018 states on page e1774, the initial follow-up for the “entire cohort was 4.6 years and the average time of follow-up was 6.8 +/- 2.24 years”. Then later in the paper, Gardner et al, states on page e1775, that indeed, “that prior TBI is associated with an increased risk of being diagnosed with PD over up to 12 years of follow-up…” Thus, the results were not intuitive, and we believed that the 12-year follow-up referred to any TBI rather than mTBI. However, after additional scanning of the text, we found that the 12-year follow-up actually referred to the 56% increased risk of being diagnosed with PD with mTBI. Therefore, we thank the reviewer for bringing this to our attention and consequently we revised our sentence accordingly (Abstract p 2, line 31, Intro p. 3, line 57).

3. The new sentence, “In summary, it is uncertain whether global or specific cognitive problems lead to mTBI-related risks for PD onset” is unclear. Are the authors proposing that post-mTBI cognition is the cause for PD onset? Please revise for clarity.

Author’s Response

Thank you for highlighting the fact that this sentence is unclear. We only wanted to state that the ambiguities in the literature make it difficult to identify a particular characteristic of cognitive functioning that might be used as a prodromal cognitive pre-motor phenotype of mTBI-related risks for PD (p. 4, line 82). We were also concerned that assessing global cognitive functioning (for example) as measured by the Mini Mental State Exam might miss more specific domains of functioning that are problematic for some patients.

We deleted this sentence and the next few words in the next sentence with the following replacement p. 4, lines 82-93 … “These ambiguities in the literature make it impossible to identify a particular domain of cognitive functioning that might be used as a prodromal phenotype of mTBI-related risks for PD” and “As such, the gap in our understanding about cognitive problems or premature cognitive decline that may link mTBI to future risks for PD is considerable. Thus, identifying a mTBI-related cognitive characteristic common to both mTBI and PD might help close this gap.” 

We hope this revision satisfies the reviewer’s concerns.

Results

4. The normative table was cut-off and was unable to be reviewed.

Author’s Response

Page 16 now reflects the proper format to incorporate a landscape table into a portrait manuscript.

5. Re: Response to Reviewer 1’s comment 4. There are quite a few studies that have examined cognitive functioning in early-stage PD. For example, Pfeiffer et al., 2014: Acta Neurol Scand 2014: 129: 307–318 DOI: 10.1111/ane.12189.

Author’s Response

We thank the reviewer for bringing this to our attention. In Pfeiffer et al study, it appears that the ‘early-stage’ PD group had mild cognitive impairment which was an exclusion criterion in our study. Moreover, we did not use the CANTAB or most other assessments as used in Pfeiffer et al 2014. So, we believe our outcomes may not correspond with this study. 

We also did not believe that we could fully use the data to uphold or contrast with our findings as the Pfeiffer study was not a match-control design, and their data could not be used in our younger sample of veterans and healthy controls in terms of similarities with our PD subjects. However, we were able to state on page 20, lines 481-487: “Indeed, in non-demented early-stage PD, Pfeiffer et al., (2014) [55] found 70% of early-stage PD with no previous history of cognitive decline met criteria for a modified mild cognitive impairment diagnosis based on the outcomes from executive functioning/working memory, attention and concentration tests while other cognitive domains were relatively spared. This finding bolsters our previous suggestion that neurocognitive evaluations need to occur early and often for both post- mTBI injury and post-PD diagnosis patients in order to best identify the trajectory affecting the cognitive domains that are most vulnerable to progression of neurodegeneration and possible conversion to cognitive impairment or dementia.

Additionally, we had previously cited Marchand et al. 2018 as this study found TMT-B was significantly predictive of future dementia in Parkinson’s disease subjects within 6-years, and we also added DeRoy et al., 2020 who conducted a match-control design with TMT-B and also found TMT-B predicted risks for dementia over a 4.3-year period in non-demented Parkinson’s [p.17-18, lines 419-429].

We agree there are other studies as we explain on p. 4 lines 82-90. Nowhere in our manuscript do we suggest that there are no other cognitive studies in early-stage PD. If we have mis-lead the reviewer with our response, that was not our intent and apologize for any confusions. 

Reviewer 1 suggested we use “…comparative data for other early-stage PD samples on the selection of cognitive tests administered”. In our previous response as shown below, we were 

Unfortunately, we could not find studies examining cognitive testing specifically in early-stage PD… other more advanced stages, yes, but not early-stage. If compared to more advanced stages, then we believe it would add confusion to our study goals and data interpretations.

strictly speaking in terms of the specific tests and primary outcomes we used were reported mostly in more advanced stages of PD, which, of course would not be an appropriate comparison for our sample. Other pertinent studies are shown in Table 1 along with the normative data to show qualitative comparisons. Overall, we apologize for being unclear about the previous response to Reviewer 1. We did not intend to reflect the actual text in the manuscript, but this statement was more of a talking point - we could not find any text in the manuscript where we stated that there were no other cognitive studies in early-stage PD. 

7. PLOS authors have the option to publish the peer review history of their article (what does this mean?). If published, this will include your full peer review and any attached files.

Do you want your identity to be public for this peer review? For information about this choice, including consent withdrawal, please see our Privacy Policy.

Reviewer #1: No

Reviewer #2: No

---

## [Editor Report · Decision Letter 2]

7 Oct 2021

Premature cognitive decline in specific domains found in young veterans with mTBI

coincide with elder normative scores and advanced-age subjects with early-stage Parkinson’s disease

PONE-D-21-11322R2

Dear Dr. Nejtek,

We’re pleased to inform you that your manuscript has been judged scientifically suitable for publication and will be formally accepted for publication once it meets all outstanding technical requirements.

Kind regards,

Firas H Kobeissy, PhD

Academic Editor

PLOS ONE
---

## [Editor Report · Acceptance letter]

22 Oct 2021

PONE-D-21-11322R2 

Premature cognitive decline in specific domains found in young veterans with mTBI
coincide with elder normative scores and advanced-age subjects with early-stage Parkinson’s disease 

Dear Dr. Nejtek:

I'm pleased to inform you that your manuscript has been deemed suitable for publication in PLOS ONE. Congratulations! Your manuscript is now with our production department. 

Kind regards, 

on behalf of

Dr. Firas H Kobeissy 

Academic Editor

PLOS ONE